# Comparative Safety of Bevacizumab, Ranibizumab, and Aflibercept for Treatment of Neovascular Age-Related Macular Degeneration (AMD): A Systematic Review and Network Meta-Analysis of Direct Comparative Studies

**DOI:** 10.3390/jcm9051522

**Published:** 2020-05-18

**Authors:** Anna A. Plyukhova, Maria V. Budzinskaya, Kirill M. Starostin, Robert Rejdak, Claudio Bucolo, Michele Reibaldi, Mario D. Toro

**Affiliations:** 1Federal State Budget Scientific Research Institute “Scientific Research Institute of Eye Diseases”, 119021 Moscow, Russia; m_budzinskaya@mail.ru; 2Medical Affairs, Sanofi-Aventis SA, 125009 Moscow, Russia; kirill_ms@yahoo.com; 3Department of General Ophthalmology with Pediatric Service, Medical University of Lublin, 20079 Lublin, Poland; robertrejdak@yahoo.com (R.R.); toro.mario@email.it (M.D.T.); 4Department of Biomedical and Biotechnological Sciences, School of Medicine, University of Catania, 95123 Catania, Italy; bucocla@unict.it; 5Department of Ophthalmology, University of Turin, 10126 Turin, Italy; 6Faculty of Medicine, Collegium Medicum Cardinal Stefan Wyszyński University, 01815 Warsaw, Poland

**Keywords:** ranibizumab, bevacizumab, aflibercept, anti-vascular endothelial growth factor, neovascular age-related macular degeneration, meta-analysis, randomized controlled trials

## Abstract

Background: Since the efficacy of ranibizumab (RBZ), bevacizumab (BVZ) and aflibercept (AFB) is comparable in neovascular age-related macular degeneration (AMD), we conducted a systematic review and meta-analysis to evaluate the long-term safety profiles of these agents, including ocular safety. Methods: Systematic review identifying randomized controlled trials (RCTs) comparing RBZ, BVZ and AFB directly published before March 2019. Serious ocular adverse events (SOAE) of special interest were endophthalmitis, pseudo-endophthalmitis, retinal pigment epithelium tear and newly identified macular atrophy. Results: Thirteen RCTs selected for meta-analysis (4952 patients, 8723 people-years follow-up): 10 compared RBZ vs. BVZ and three RBZ vs. AFB. There were no significant differences in almost all adverse events (systemic and ocular) between BVZ, RBZ and AFB in up to two years’ follow-up. Macular atrophy was reported heterogeneously and not reported as SOAE in most trials. Conclusions: Direct comparison of RBZ, BVZ and AFB safety profiles in the RCT network meta-analytical setting have not revealed a consistent benefit of these three commonly used anti-vascular endothelial growth factor (anti-VEGF) agents in AMD. Network model ranking highlighted potential benefits of RBZ in terms of a systemic safety profile; however, this appears a hypothesis rather than a conclusion. Newly identified macular atrophy is underestimated in RCTs—future real-world data should be focused on SOAE.

## 1. Introduction

Age-related macular degeneration (AMD) is the leading cause of visual disability among patients over 60 years. The most prevalent major advanced form of the disease, the “dry” (atrophic) form, is characterized by a slow, progressive dysfunction of the retinal pigment epithelium, photoreceptor loss and retinal degeneration; on the contrary, the “wet” (neovascular) form is less frequent but responsible for 90% of acute blindness due to AMD [1]. Wet AMD is associated with a vascular endothelial growth factor (VEGF) increase and blood vessels growing de novo; therefore, anti-VEGF agents can be injected in the eye to inhibit neovascularization and prevent the gradual loss of vision. Anti-VEGF treatment showed significant benefit over placebo, being now a pillar of neovascular AMD treatment, providing over a 90% chance of stabilizing or increasing vision after two years of treatment [2]. Currently, there are three main anti-VEGF agents available in clinical practice to manage wet AMD: ranibizumab (RBZ), aflibercept (AFB) and bevacizumab (BVZ), the latter used off-label. Characteristics of these three agents are summarized in Table 1.

Even though the pharmacological targets of these three drugs is the same, the structures are different; therefore, the mechanism of action and the pharmacokinetics profile may differ [4,5], with an impact in terms of the risk/benefit ratio. Further, another remarkable difference is the cost of these products. Roughly BVZ is about 40 times cheaper than RBZ and AFB, (50, 1800 and 2000 USD for BVZ, AFB and RBZ, respectively; www.aao.org available data). Appropriate drug use, even off-label use, is always a doctor’s decision, unless a specific law in some country limits the off-label prescription. In this context, the intraocular off-label use of BVZ is probably driven by the cost and not by clinical benefit [6].

As regards the efficacy, there are a several randomized controlled trials (RCT) studies and meta-analyses collecting the results of these three agents showing benefit over sham injections. No significant difference is revealed in terms of impact on visual acuity in neovascular age-related macular degeneration (nAMD), though the number of injections needed to achieve these clinical effects was reported lower for AFB. For instance, in nAMD [7], AFB and RBZ had comparable efficacies for best-corrected visual acuity (BCVA) and central macular thickness (CMT); however, these results were achieved with less injections (about five) of AFB (vs. RBZ) over two years. BVZ and RBZ had equivalent efficacies for BCVA as well, and the only difference constituted a greater RBZ reduction in CMT compared to BVZ [7].

Recently, a review compared the effectiveness and safety of these three anti-VEGF drugs in preserving and improving vision and quality of life using network meta-analysis methods [8] with no signals of differences in overall safety between the three antiangiogenic drugs, even though the authors claimed that their estimates are imprecise for cardiovascular events and death [8]. Noteworthy, (1) ocular events were not in scope, and (2) this analysis compared anti-VEGF vs. different types of control, not only head-to-head studies. The current network meta-analysis was focused on comparing both systemic and ocular adverse events of these three molecules in large-scale (at least n = 50 per group), long follow-up (at least one year) head-to-head trials. A retinal pigment epithelium (RPE) tear and macular atrophy were adverse events of special interest in anti-VEGF treatments, potentially considered as a serious adverse event associated with this treatment and leading to anti-VEGF treatment withdrawals (a decision which may be seen as inappropriate). So, these ocular adverse events are appealing from a scientific and clinical point of view [9,10]. Among other ocular events, we highlighted endophthalmitis as one of the most dangerous local complications associated with intravitreal injections [11]. Finally, this work focused on direct head-to-head comparisons without indirect comparisons through other interventions to avoid an additional source of bias and heterogeneity.

## 2. Materials and Methods

This systematic review was conducted and reported in accordance with the Preferred Reporting Items for Systematic Reviews and Meta-Analyses (PRISMA) guideline for meta-analyses (see the PRISMA network meta-analysis (NMA) checklist, Appendix A).

### 2.1. Search Strategy

Study search was conducted using PubMed, Excerpta Medica data BASE (EMBASE (all via OVID Medline)), and the Cochrane Library from their inception until 1 July 2018. Available systematic reviews were also analyzed to double-check comprehensiveness of the search conducted de novo. Search terms embracing all PICOTS (patient-intervention-comparison-outcome-timing-setting/study design) aspects were developed to construct the optimal search strategy and identify patients diagnosed with AMD and treated with BVZ, RBZ or AFB in direct comparative head-to-head trials. Included trials are provided in Appendix A.

### 2.2. Selection of Studies

Search results were consolidated with all relevant abstracts screened and selected independently by two reviewers (A.A.P. and K.M.S.). In case of any uncertainty, full texts were additionally screened. Then, full-text manuscripts of preselected abstracts were retrieved and assessed based on eligibility criteria. All discrepancies in expert assessments were clarified in discussion. Reasons for study exclusions were aggregated and presented below (Table 2).

Additional note: During our work, unpublished Intravitreal Aflibercept for Neovascular Polypoidal Choroidal Vasculopathy clinical trial (RIVAL) study (NCT02092532) data on macular atrophy events became available, so we included it into the analysis as well.

### 2.3. Inclusion and Exclusion Criteria

Criteria of study inclusion in this systematic review were developed and defined as follows (PICOTS approach):(P) Study population: patients diagnosed with neovascular AMD(I) Interventions of interest: intravitreal BVZ, RBZ or AFB(C) Treatment comparisons: BVZ, RBZ and AFB(O) Outcomes of interest: any safety data reported (death, systemic and ocular serious adverse events)(T) Minimal follow-up period: 1 year(S) Studies were designed as RCTs. Studies published in English were eligible for inclusion in the review. Only studies that provided sufficient data of safety outcomes with at least 1-year follow-up and compared the safety of the following interventions: 1.25 mg BVZ or 0.5 mg RBZ monthly, as needed (pro re nata) or a treat-and-extend regimen, 2 mg AFB every 2 months or treat-and-extend regimen after 3 initial monthly doses (2q8), were included for network meta-analysis (NMA).

Secondary analysis studies, reviews, were excluded. Studies on macular dystrophy/neovascularization of any other origin and studies with no control, sham control or active control other than BVZ, RBZ, AFB were excluded as well.

### 2.4. Data Extraction

Authors, year of publication, baseline characteristics of included studies and outcomes of interest were independently extracted by two reviewers (A.A.P. and K.M.S.). Any disagreement was resolved by discussion. We also contacted study sponsors and authors of the articles to request full information we were interested in.

### 2.5. Outcomes of Interest

Primary outcomes were the (1) proportion of dead patients; (2) patients having experienced at least one serious systemic event; (3) at least one cardiovascular event: (3a) atherothrombotic event or (3b) venous thrombotic event and (4) at least one serious ocular event: (4a) endophthalmitis, (4b) pseudo-endophthalmitis, (4c) retinal epithelium tear or (4d) macular atrophy during follow-up.

### 2.6. Risk of Bias Assessment

Two authors independently assessed and reported the methodological risk of bias of the included studies using the relevant Cochrane collaboration’s tool, assessing randomized trials according to the following domains of potential bias: randomization process, deviations from intended interventions, missing outcome data, measurement of the outcome, selection of the reported result and overall bias (Appendix A). Systemic serious adverse events (SAE) are considered as the most critical, being related to a solid endpoint, and it has a clear definition according to good clinical practice, so generally, it is supposed to be homogenously recognized across studies; however, we need to highlight the potential risk of bias in terms of its reporting based on the assessment of potential bias risk (Section “Incomplete outcome data”, Appendix A).

### 2.7. Statistical Analysis

Mortality rate and other adverse events were presented as frequencies and percentages. We contacted sponsors or authors for providing aggregated data when data were not reported. Results of quantitative pair-wise analysis was presented using forest plots. R studio software (version 1.1.453) was used to perform meta-analysis and all statistical procedures. A two-sided test with *p*-values less than 0.05 were considered as statistically significant. Fixed effects and random effects meta-analytical model of Mantel-Haenszel was performed (based on incidence rate ratio) with a DerSimonian-Laird estimator for tau2 in frame of the R package “metaphor” (Viechtbauer 2010) for meta-analysis. Graph-theoretical method was applied for network meta-analysis using “netmeta” R package (equivalent to the frequentist approach) [12]. Trials with zero event arms were incorporated into the analysis to include all relevant data, regardless of the effect measure chosen [13]. Only newly identified macular atrophy zero event cases were not included, since atrophy usually are not reported as ocular SAE, and this is a part of further discussions.

### 2.8. Sensitivity Analysis

We planned to conduct sensitivity analyses to assess the impact of studies graded as having a high risk of bias on any parameter, unpublished data only or based on withdrawal rate. After assessing the data collected, we determined these analyses were not needed, since we did not identify highly heterogenous data that were worth a special quantitative analysis.

## 3. Results

Study selection flow chart of this systematic review is reported in Figure 1. We identified 4041 items from PubMed, EMBASE and the Cochrane Library. Study criteria exclusion is provided in Table 2. Among 33 eligible RCT studies, 12 included safety information from trials, while the others were classified as post hoc/ad hoc analyses or having no safety information. Four studies were excluded from quantitative synthesis: all these studies were one-year follow-up reports with two-year follow-up reports existing in parallel, so the latter were included only to analyze the maximal available duration of treatment (VIEW 2, CATT 2, IVAN 2 and LUCAS 2 studies).

### 3.1. Characteristics of Included Studies and Quality Assessment

Basic characteristics of the RCT included and the safety data availability there are given in the Table 3 and Table 4, accordingly.

### 3.2. Risk of Bias Assessment

Details on the risk of bias assessment are presented as Appendix A, attached.

### 3.3. Safety Analysis

#### 3.3.1. BVZ vs. RBZ

Incidence rate ratio of death, certain systemic SAE and total and ocular SAE in RBZ and BVZ groups did not differ significantly. Only systemic SAE were significantly higher in the BVZ group (*p* = 0.035 in the pair-wise meta-analytical model). No relevant heterogeneity was identified among these trials (Appendix A).

#### 3.3.2. AFB vs. RBZ

Since RIVAL data were not published fully when performing this meta-analysis, we included all available safety data regarding this pair-wise comparison based on VIEW1,2 studies, plus those received from a RIVAL study sponsor. Heterogeneity was suspected in newly identified atrophy between these two studies in spite of the fact that, formally, it was insignificant as per appropriate statistics (I^2^ = 19%, df = 1, *p* = 0.27), as also shown in a funnel plot in Appendix A. The reason for this suspicion was that, in the RIVAL study, newly identified atrophy was identified in > 20% subjects in each group, whereas in VIEW1,2, this complication was identified in less than one percent in both groups.

#### 3.3.3. AFB vs. BVZ 

No direct trials were identified. Therefore, only adjusted indirect comparison is available based on the network meta-analysis (see Figure 2 and Figure 3 and Appendix A). Based on the random effects model, there was no significant difference in the safety profiles of BVZ and AFB.

#### 3.3.4. Synthetic Comparison of Three Options

The network meta-analytic model was developed based on BVZ vs. RBZ and BVZ vs. AFB direct comparison. In the frame of this model, AFB vs. BVZ were compared indirectly. Based on this network meta-analytic model, the incidence rate ratio was for safety events and presented in a forest plot (Figure 3).

Besides the significant difference between RBZ and BVZ in the systemic SAE rate, no other significant difference was revealed in any BVZ vs. RBZ, RBZ vs. AFB or RBZ vs. AFB comparisons. Venous thrombotic events and pseudo-endophthalmitis events were not included in the network meta-analytic model, being available only in the RBZ vs. BVZ comparison. Cardiovascular diseases (CVD) death in the RBZ vs. AFB comparison was near the significance borderline. Considering the *p*-score (SUCRA analogue in the frequentist network model), RBZ seems to have the potential of being superior over BVZ and AFB in terms of general surrogate safety parameters (dropout and death), over AFL in terms of CVD death and over BVZ in terms of systemic SAE (Figure 4). Heterogeneity in newly identified macular atrophy was still formally insignificant (I^2^ = 42.8%, df = 2, *p* = 0.17); however, qualitatively, we consider these data as heterogeneous. Other endpoints were considered homogenous.

## 4. Discussion

RBZ and AFB are commonly anti-VEGF drugs used in clinical practice to manage AMD. At the same time, other treatment off-label options are available, such as BVZ, this latter likely due also to cost. BVZ is the most commonly used option alternative to on-label anti-VEGF drugs. Even though these drugs (RBZ, AFB and BVZ) have the same pharmacological target, they have not the same structure and are not equivalent. It is noteworthy that the AFB/VEGFA complex is characterized by electrostatic stabilization with a high association rate and high stabilizing electrostatic energy, whereas RBZ and BVZ complexes were stabilized by Van der Waals energy term [5]. RBZ is supposed to have a lower dissociation rate with lower conformational fluctuations of the RBZ/VEGFA complex and a higher number of contacts and hydrogen bonds in comparison to BVZ and AFB. [5]. According to the pharmacokinetics study by Christoforidis and co-workers (2017), intravitreal half-lives for these three molecules in the case of intraocular injections were 3.60 days for BVZ, 2.73 days for RBZ and 2.44 days for AFB. Serum levels were highest for BVZ (area under the curve (AUC) = 132 ng/mL) as compared to both RBZ (AUC = 3 ng/mL) and AFB (AUC = 44 ng/mL). In terms of the systemic exposure duration, BVZ was the most prolonged (Tmax = 4 days) compared to RBZ (Tmax = 1 day) and AFB (Tmax = 2 days) as well. However, all agents were primarily excreted through the renal and mononuclear phagocyte systems. BVZ was found with high levels in the liver, heart and distal femur bones [5]. Based on that knowledge, it is of value to compare directly the safety profiles of these three anti-VEGF treatments. So, we analyzed the extended safety profile outcomes with not only systemic but, also, ocular SAE, including de novo macular atrophy and retinal pigment epithelium tears, since they are characterized with constantly increasing scientific interest and clinical importance. At the same time, we narrowed the search strategy to direct comparative trials to get more homogenous comparative arms under the network meta-analysis.

Systemic adverse events that proved to be significantly higher in BVZ than in RBZ (*p* = 0.035 in pair-wise RBZ vs. BVZ meta-analytical model) is mostly due to the The Comparisons of Age-Related Macular Degeneration Treatments Trials (CATT2) study, and it is supposed to be related to different pharmacokinetics profiles. Noteworthy, recent pharmacokinetic (PK) clinical studies carried out in humans are in accordance with pre-clinical data. Thus, in 2014, Avery et al. published comparative PK data showing that systemic exposure to AFB after intravitreal injection was nine-fold higher than in RBZ, and for BVZ, this parameter was much more—35-fold higher than that in RBZ [14]. In 2017, these data were replicated by the same author group in a large sample [15]. CVD death differences between AFB and RBZ in our study did not reach statistical significance; nevertheless, in VIEW 1,2, considering it separately, it was near this level (pair-wise Fisher’s exact test provided *p* = 0.052). RIVAL study changed almost nothing in terms of this endpoint, since there were no CVD deaths registered. So, in the meta-analytical random effect model, the CVD death rate difference was still insignificant with *p* = 0.0625 (fixed effect model, *p* = 0.0559). An interesting finding is that, in the RIVAL study, AFL was associated with 2.03 higher death incidences than RBZ; however, this difference was insignificant as well. Moreover, the AFB group ratio of subjects with atherothromboembolic anamnesis was 1.7 higher than in the RBZ group (18.0% vs. 10.6%), though considered as insignificant (as per the chi square test, *p* = 0.075). Since no CVD death was reported, it is impossible to explain the overall death difference by atherothrombotic anamnesis adjustment. This probably will be reflected in RIVAL publications or may be considered statistically accidental.

The other adverse events did not differ significantly, both systemic adverse events and ocular adverse events. Further network ranking, a method that is free from the classic “*p* = 0.05/0.01” level of significance, showed that RBZ had a greater safety potential in terms systemic events compared to AFB, whereas ocular adverse events had no tangible difference even in the network ranking, and therefore, the ocular safety profile of these three anti-VEGF drugs seems to be the same. It is important to understand that p-score ranking measures mean the extent of certainty that a treatment is better than others, albeit telling us nothing definitely [16]. So, readers should interpret with caution the ranking of treatments and should be careful in the use of rankings to guide practices. Nevertheless, the question is whether this systemic potential difference is in-line with other studies, and in case this difference exists, what are the most probable type of events? It is also important to understand what the main contributor to this SAE difference is. Finally, another question is whether we can confirm that the ocular safety profile may be stated as comparable for these three interventions.

Comparing our systemic adverse events data with other analytical reports, we noticed contradictory results. For instance, Zhang and co-worker’s meta-analysis found low incidences of venous thrombotic events with RNB, but the risk of atherothrombotic events did not differ [17]. Another meta-analysis that included not only AMD but, also, Diabetic macular edema (DME) and retinal vein occlusion (RVO) showed that BVZ increased the risk of venous thromboembolic events [16]. Several meta-analyses of AMD trials found increased risks of gastrointestinal adverse events with BVZ vs. RBZ [18,19,20]. At the same time, one of the Cochrane meta-analyses had no difference in terms of the SAE [21]. Based on the above considerations, and taking into account potential imbalances in baseline status in some studies (e.g., LUCAS) and the statistical dispersion in the synthetic quantitative analysis with enhanced degrees of freedom, we believe that questions regarding anti-VEGF systemic and ocular safety profiles should not be solved in the frame of RCT analysis only. This hypothesis of RBZ having potential benefits in terms of systemic safety should be thoroughly tested in a real-world data setting. The key questions there would be comparable patient profiles and the power of the study sample, and the latter is of the utmost importance when taking into account already published real-world data studies. For example, in a recent real-world data analysis of comparative retrospective cases of AFB and RBZ, there were no adverse events recorded in any group, which may be explained by (1) a small sample size or (2) tendency not to report adverse events in observational studies [22]. Noteworthy, RIVAL study results are expected to be published fully, which may give us new data in RCT settings comparing AFB and RBZ (NCT02130024).

Ocular adverse events are another safety issue that attracts clinical attention. We decided to focus on those that are most common or valuable in terms of clinical impact. We also considered its usual rare frequency and included the total ocular SAE as well. Nevertheless, we failed to reveal any difference in ocular events. As far as we can see, even in the meta-analysis, the rareness of these events made it impossible to identify even existing differences in the case that it does takes place. We had a mean ocular SAE number in RNB of 30 events/1000 people-year (CATT2, IVAN2, GEFAL and VIEW1,2); BVZ 26 events/1000 people-year (CATT2, IVAN2 and GEFAL) and AFB 20 events/people-year (VIEW1,2). To have a significant difference in a setting with n = 500 in one trial, the difference should be at least 20 vs. 40 events, which means twice, not 1.5,-fold difference (Fisher’s exact test). Since some trials did not report a total ocular SAE number (LUCAS2 and BRAMD) or reported zero (Biswas and MANTA), our analysis is not comprehensive (potential publication bias).

Endophthalmitis does not seem to be associated to specific anti-VEGF treatments, whereas noncompliance with recognized hygiene standards and poor aseptic techniques are accounted responsible for the endophthalmitis outbreaks reported. Even such simple measurements as masks and silence are proven to be truly important, especially taking into account that, usually, the cause of endophthalmitis after intravitreal injections turns out to be a *Streptococcus* species, which comprise more than 40% of culturable adult salivary flora [11,23]. Our search strategy was not designed to analyze specifically the endophthalmitis rate, as we had a one-year minimum follow-up to focus on the long-term safety profile, whereas endophthalmitis is usually reported within the first several weeks [23,24]. Etiology of sterile endophthalmitis (pseudo-endophthalmitis), independently of the administered drug, remains uncertain, and its multifactorial origin cannot be discarded [25]. However, the endophthalmitis rate difference in VIEW1,2 was statistically significant (*p* = 0.03, Fisher’s exact test); in huge datasets, it was shown that there is no significant difference in endophthalmitis among RBZ, BVZ and AFB [26].

When it comes to pigment epithelium tears, there have been a large number of reports describing this complication after intravitreal BVZ, RBZ and AFB therapy, almost always in eyes with vascularized pigment epithelial detachments (PED) and neovascular AMD. The pathogenesis of RPE tear formations continues to remain controversial; at the same time, it seems to be related to the size of the presenting PED at the baseline, stress on the RPE and the anti-VEGF treatment [27]. This complication may occur spontaneously or during the anti-VEGF treatment, which is considered as a risk factor, but the true extent of the treatment’s contribution to the natural course of this complication is still unclear [27,28].

Finally, macular atrophy is reported in RCT rarely, and this is probably the most intricating adverse event. First of all, it is critical from a clinical point of view. Secondly, atrophy developed de novo may be considered as a direct consequence of the anti-VEGF treatment and transforming wet AMD into dry [29]. Surprisingly, in RCTs, this adverse event is reported rarely (no data in five RCTs of nine; see Table 4). However, this event takes time to develop and may be associated with the drying regimen of anti-VEGF treatments; the most relevant explanation appears to be that a macular atrophy definition and method of diagnosis is not standardized yet and may vary across studies, so its detection and investigation is hampered by the absence of consensus on this clinical phenomenon, with inevitable heterogeneity reporting, which we revealed among the studies. RIVAL unpublished study, IVAN2 and CATT2 had about 20%–30% patients in two years with retinal atrophy developed de novo, whereas other RCTs did not report this event (LUCAS2, Gefal, BRAMD, Biswas and MANTA) or reported a few events per group (VIEW1,2). At the same time, a SEVEN UP study reported that 90% of eyes that had received seven years of treatment had atrophy involving the foveal center; nevertheless, this study reported also that 23% of their patients had visual acuity ≥ 6/12, which looks inconsistent [29]. So, to perform further analyses of ocular adverse events, observational data and real-world data should be taken into account.

### 4.1. Limitations

Only the three-most common anti-VEGF treatments were included; no pegaptanib, no ziv-AFB or additional compounds that may be co-administrated with anti-VEGF molecules were included (e.g., fovista aptamer—NCT01940887 or sirolimus—NCT02732899). Only AMD was under investigation without DME and RVO. Monthly and treat-to-extent regimens were aggregated and analyzed together, presuming a comparable safety profile of these treatment regimens. Sample size limitation of n = 50 per group led to a Subramanian study exclusion, which, however, does not seem to influence the results, since no systemic or ocular serious adverse events were reported in one year [30]. No switching was taken into account. In this connection, there was an interesting study in which the results were expected—a TIDE AMD comparing a regimen of monthly monotherapy of RBZ with planned switching to AFB (NCT02257632). Short-term studies were excluded, since long-term follow-up and systemic events were prioritized. In this connection, our analysis should not be taken as a comprehensive short-term ocular events analysis. Thus, we had to exclude the UNRAVEL study (NCT01988662) due to a short follow-up period (three months); however, no death was recorded there, and only three SAE were reported on RBZ (among two ocular hemorrhages only) and five on AFB (one ocular hemorrhage).

### 4.2. Strengths

Only direct comparisons were considered to avoid heterogeneity and the cumulative incidence rate design of the analysis without dividing the results per years of follow-up. All zero event arms were included; new RIVAL data was included. The p-score network ranking was provided, and both random and fixed effects models were reported.

## 5. Conclusions

Based on available direct comparative RCTs of RBZ, AFB and BVZ, it may be concluded that RBZ appears to have some benefits in terms of systemic adverse events, whereas in terms of ocular adverse events, no statement regarding the superiority of any of these three agents may be provided. At the same time, since the available studies were underpowered to figure out the safety difference, especially rare ocular adverse events, the former statement regarding systemic adverse events should be taken as a hypothesis for further studies, whereas no difference in ocular adverse events means only that insufficient data are available so far to draw a final conclusion. Therefore, both systemic and ocular safety profiles of anti-VEGF agents should be further evaluated in large-scale real-world data studies.

## Figures and Tables

**Figure 1 jcm-09-01522-f001:**
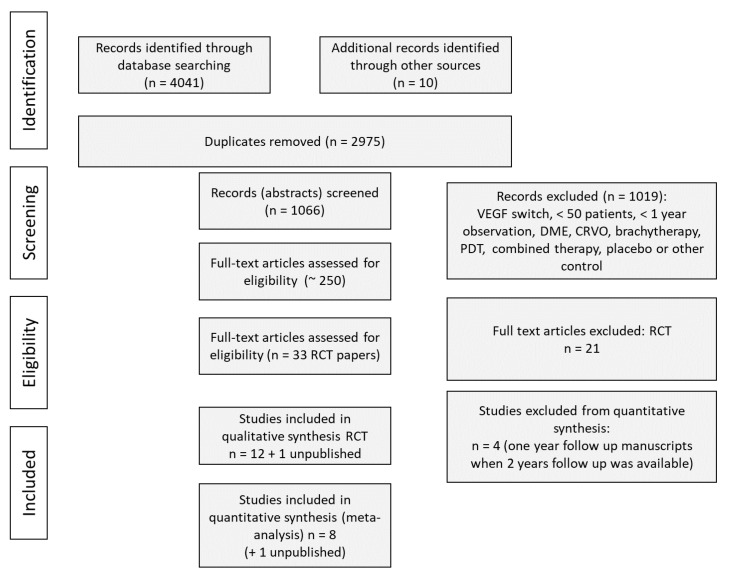
Study flow diagram Preferred Reporting Items for Systematic Reviews and Meta-Analyses (PRISMA) showing number of trials identified, included, excluded and reason for exclusion.* -.

**Figure 2 jcm-09-01522-f002:**
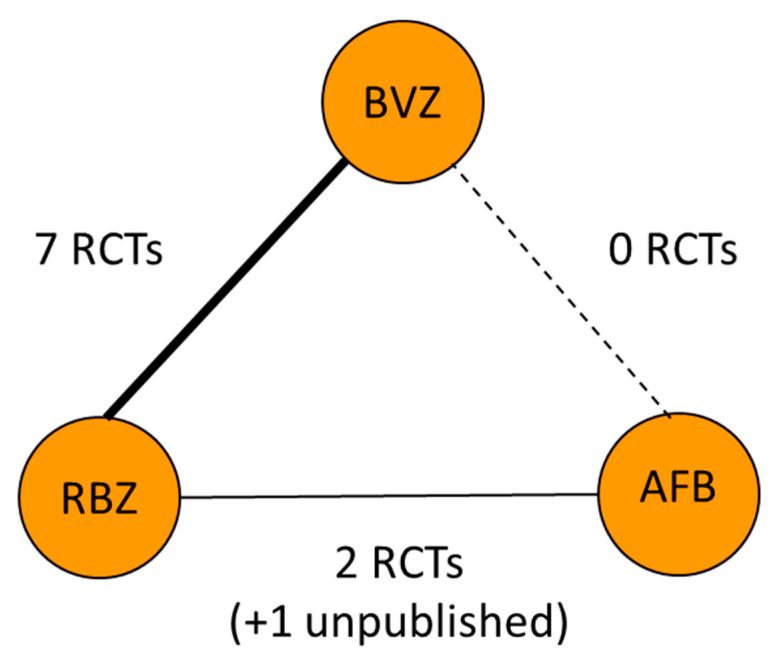
Node graph of the network meta-analytic model developed. Richness of the line means a number of studies available. Solid line—head-to-head RCTs are available. Dotted line means no head-to-head RCTs are available and an indirect comparison in the frame of the network meta-analysis is performed.

**Figure 3 jcm-09-01522-f003:**
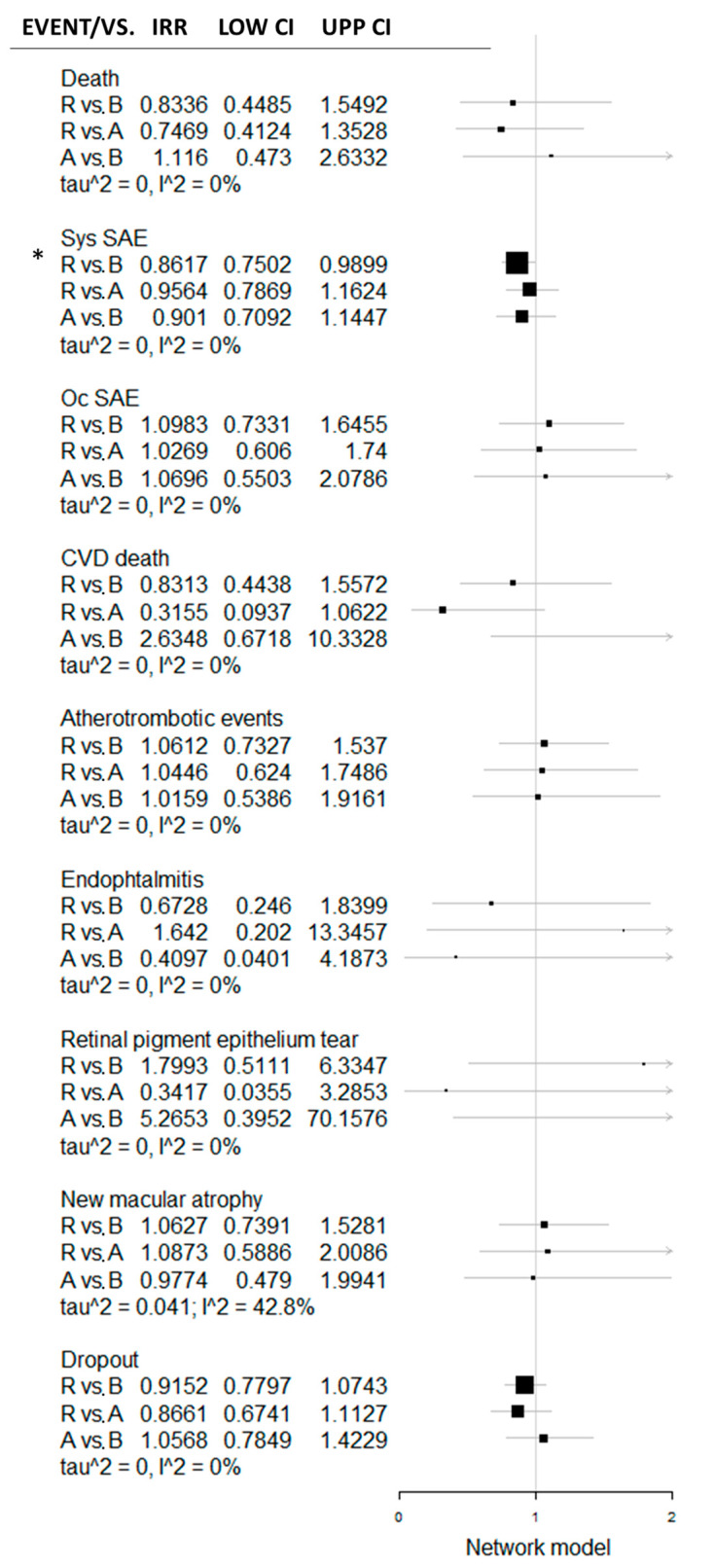
Forest plot with direct and indirect comparison of RBZ, BVZ and AFB in the frame of the network meta-analytic model. Systemic serious adverse event (SAE) rate differs significantly between BVZ and RBZ, whereas the other safety parameters’ difference was recognized as statistically insignificant, both direct and indirect. *—significantly higher systemic SAE rate in BVZ vs. RBZ. R-ranibizumab; B-bevacizumab; A-aflibercept; CVD-cardiovascular diseases, Oc SAE–ocular serious adverse events; Sys SAE-systemic serious adverse events; tau2-between-study variance; I^2^–percentage of variance related to between-study heterogenicity; LOW CI–lower limit of confidence interval; UPP CI–upper limit of confidence interval; IRR–incidence rate ratio; vs.–versus.

**Figure 4 jcm-09-01522-f004:**
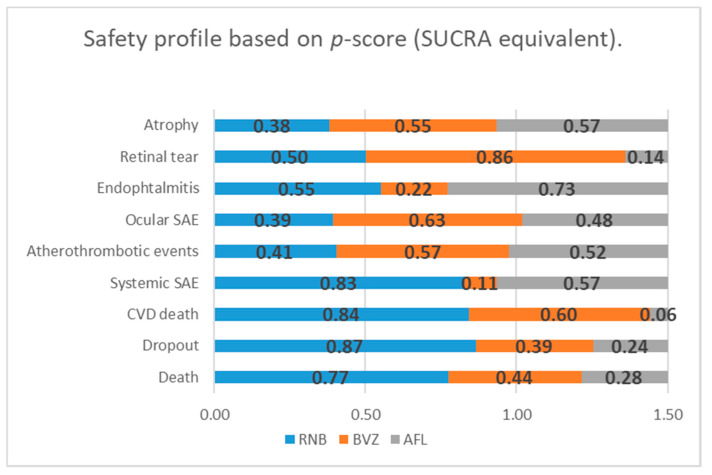
Safety profile of three anti-VEGF interventions based on the *p*-score values estimated using the frequentist network meta-analytic model. SUCRA - surface under the cumulative ranking curve. *P*-scores are similar to SUCRA values in the Bayesian network model and the means probability of being the best treatment option.

**Table 1 jcm-09-01522-t001:** Comparative characteristics of BVZ, RBZ and AFB, adapted from [3]. AMD: age-related macular degeneration.

	Bevacizumab (BVZ)	Ranibizumab (RBZ)	Aflibercept (AFB)
**Manufacturer**	Avastin; Genetech, South San Francisco, CA, USA	Lucentis; Genetech, South San Francisco, CA, USA	Eylea; Regeneron Pharmaceuticals, Tarrytown, NY, USA
**Type of molecule**	Full-size recombinant humanized IgG1 kappa monoclonal antibody	Fab fragment of a recombinant humanized IgG1 kappa isotype murine monoclonal antibody	Fusion protein of the second Ig domain of human vascular endothelial growth factor receptor 1 (VEGFR-1) and the third Ig-binding domain of human VEGFR-2 with the constant fragment crystallizable portion of the human IgG1
**Molecular weight**	149 kDa	48 kDa	115 kDa
**Picture**	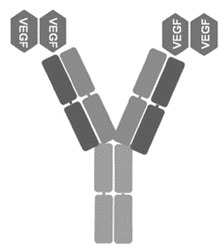	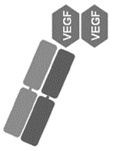	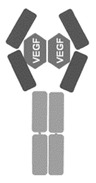
**Comments**	N-glycosylated in its Fc region	Not glycosylated	NA
**Production**	Mammalian cell lines CHO DP-12	*Escherichia coli* cells, recombinant DNA technology	Hamster ovary cells
**Receptor-ligand interaction**	Against all isoforms of VEGF-A	Against all isoforms of VEGF-A	Binds to all isoforms of VEGF-A (higher affinity than BVZ and RBZ); also binding to VEGF-B and Placental Growth Factor (PIGF).
**Authorization in the USA**	FDA in 2005, colorectal and breast cancers, used in AMD off-label	FDA in 2006, AMD	FDA—2011, AMD

**Table 2 jcm-09-01522-t002:** Reasons for abstract exclusions.

Reason for Exclusion	Number of Articles Excluded
All references identified	4043
Duplicates	2975
Unique references	1068
Excluded unique references	
Reviews	22
Brachytherapy	5
Combined anti-VEGF treatment	8
Central retinal vein occlusion (CRVO)	15
Diabetes macular edema	61
Less than one-year follow-up	31
Mixed diagnosis AMD and macular degeneration (MD) caused diabetic macular edema (DME)/ Polypoidal choroidal vasculopathy (PCV)	7
Photodynamic therapy(PDT)	5
Switch	28
Other reasons for exclusion (no direct comparison, secondary analysis, observational studies, etc.)	747
Not published as a full manuscript (no safety data available)	13
Included unique references	
RCT	33 (12 unique studies)

**Table 3 jcm-09-01522-t003:** Basic characteristics of the RCT included in the analysis.

Author Study	Treatment Groups	Regimen Doses	Following Months	Number of Patients: BVZ/RBZ or AFB/RBZ	Age, Years
CATT 1 (2011)	BVZ OR RBZ	0.50 mg/in 0.05 mL RBZ or 1.25 mg/0.05 mL BVZ cont/discont	12 m	586/599	79.7/78.8
CATT 2	BVZ OR RBZ	0.50 mg/in 0.05 mL RBZ or 1.25 mg/0.05 mL BVZ cont/discont	24 m	586/599	79.7/78.8
IVAN 2012	BVZ OR RBZ	1.1.1.1 cont/discont regimen BRZ (0.5 mg) or BVZ (1.25 mg)	12 m	296/314	77.8/77.7
IVAN 2013	BVZ OR RBZ	1.1.1.1 cont/discont regimen BRZ (0.5 mg) or BVZ (1.25 mg)	24 m	296/314	77.8/77.7
GEFAL 2013	BVZ OR RBZ	BVZ 1.25 mg or RBZ 0.50 mg in 0.05 mL of solution following treat-and-extend protocol	12 m	191/183	79.6/78.7
BRAMD 2016	BVZ OR RBZ	Monthly 1.25 mg BVZ or 0.5 mg RBZ	12 m	161/166	79/78
LUCAS 2015	BVZ OR RBZ	RBZ 0.5 mg or BVZ 1.25 mg following a treat-and-extend protocol	24 m	213/218	62/78
MANTA 2013	BVZ OR RBZ	RBZ 0.5 mg or BVZ 1.25 mg following a treat-and-extend protocol	12 m	154/163	76.7/77.6
Biswas 2011	BVZ OR RBZ	RBZ 0.5 mg or BVZ 1.25 mg monthly	18 m	50/54	64.4/63.5
VIEW 1	AFB OR RBZ	AFB 0.5 mg monthly (0.5q4), 2 mg monthly (2q4), 2 mg every 2 months after 3 initial monthly doses (2q8), or RBZ 0.5 mg monthly (Rq4)	24 m	911/304	78/78
VIEW 2			24 m	913/291	74/73
RIVAL	AFB OR RBZ	AFB 2.0 mg OR 0.5 mg RBZ in a treat-and-extend regimen	24 m	139/142	76.6/78.7

**Table 4 jcm-09-01522-t004:** Safety data availability across RCTs included into the analysis as per designed a priori outcomes (Y—data reported and N—data not available). SAE: serious adverse events.

	CATT 2	IVAN2	Gefal	BRAMD	LUCAS2	MANTA	Biswas	VIEW12	RIVAL2
patients dead	**Y**	**Y**	**Y**	**Y**	**Y**	**Y**	**Y**	**Y**	**Y**
patients with ≥ 1 systemic SAE	**Y**	**Y**	**Y**	**N**	**Y**	**Y**	**Y**	**Y**	**Y**
patients dead from cardiovascular (CV) event	**Y**	**Y**	**Y**	**N**	**Y**	**N**	**Y**	**Y**	**Y**
patients with ≥ 1 venous thrombotic events	**Y**	**Y**	**Y**	**N**	**Y**	**Y**	**Y**	**N**	**N/av**
patients with ≥ 1 atherotrombotic events	**Y**	**Y**	**Y**	**N**	**Y**	**Y**	**Y**	**Y**	**Y**
patients with ≥1 ocular SAE	**CATT1 only**	**Y**	**Y**	**N**	**N**	**Y**	**Y**	**Y**	**N/av**
endophtalmitis	**Y**	**N**	**Y**	**N**	**Y**	**Y**	**Y**	**Y**	**Y**
pseudoendophtalmitis	**Y**	**N**	**N**	**N**	**Y**	**Y**	**Y**	**N**	**N/av**
patients with retinal pigment epithelium tear	**CATT1 only**	**Y**	**N**	**N**	**Y**	**Y**	**Y**	**Y**	**N/av**
new macular atrophy	**Y**	**Y**	**N**	**N**	**N**	**N**	**N**	**Y**	**Y**
dropout rate	**Y**	**Y**	**Y**	**Y**	**Y**	**Y**	**Y**	**Y**	**Y**

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
