# Peer review of "Comparative Safety of Bevacizumab, Ranibizumab, and Aflibercept for Treatment of Neovascular Age-Related Macular Degeneration (AMD): A Systematic Review and Network Meta-Analysis of Direct Comparative Studies"

_jcm, 2020, doi:10.3390/jcm9051522_

Round 1

Reviewer 1 Report

I have read with much interest the paper: "Comparative Safety of Bevacizumab, Ranibizumab, And Aflibercept For Treatment of Neovascular Age-Related Macular Degeneration (AMD): A Systematic Review and Network Meta-Analysis of Direct Comparative Studies", which I found clear and concise.    The main finding that the authors explore is the difference in adverse events among the 3 anti-VEGFs used in neovascular AMD treatment, a theme where much debate still exists despite multiple attempts to explore it, and an important one since it can change the perception and choice of treatments by physicians.    In fact, in the model created for comparison,  it was found that "besides significant difference between RBZ and BVZ in systemic SAE rate no other significant difference revealed in either BVZ vs RBZ, RBZ vs AFB or RBZ vs AFB comparison." Which is an interesting and important finding but not totally unexpected.  Also, by using p score (SUCRA analogue in frequentist network model) the authors found that "RBZ seems to have the potential of being superior over BVZ and AFB in terms of general surrogate safety parameters (dropout, death), over AFL in terms of CVD death, over BVZ in terms of systemic SAE." This is important information, but caution must always be kept in mind and further meta-analysis on the matter with real-life world data should be conducted to replicate these findings, as the authors correctly state in Conclusions.    Point-by-point review:    Introduction: Some minor errors and better English rephrasing is needed.  Page 2 - contest instead of context. Page 3: "As regards the efficacy, there are a lot of RCT studies", perhaps "there are several, ". Page 3: "Tear of retinal pigment and geographical atrophy", better describe those as Retinal pigment epithelium tear and geographic atrophy.  However, the term "macular atrophy" would be preferable to use instead of "geographic/ geographical", as the latter is now used by consensus only in cases of cRORA (atrophy) but without a history of neovascularization. Same in page 5, line 135.    Also, in the introduction, the anti-VEGF discussion should be more focused on AMD only and comparative trials, and not extend to diabetic macular edema or venous occlusion, as the pathophysiology is entirely different compared to AMD and is out of the scope of the paper.    Methods are appropriate, only minor questions to elucidate: Page 4 line 121 - "safety of the following interventions: 1.25 mg BVZ, 0.5 mg RBZ, 2 mg AFB 121 every 2 months after 3 initial monthly doses (2q8), were included for network meta-analysis (NMA)." What were the interventions on BVZ and RBZ? Only AFB is described.  Page 5, line 143: "events were presented as count and percentage.", perhaps change to frequencies and percentages.   Results: Table 4 - some results "Y" are in color red.   It would be nice to have a table exposing and detailing the main "systemic adverse" events explored in the included studies, as this is the outcome most important.    Discussion:  Page 11, line 276: "For instance, Zhang and co-workers (ref?) meta-analysis", is sentence meant to be like this? "(ref?)" Page 11, line 292: reference 22 in manuscript, but in the bibliography is ref 23 the real-life study.  Page 12, line 319: " Several authors showed that almost all cases of RPE tear identified after anti-VEGF therapy was associated with a baseline vascularized PED. So vascularized pigment epithelial detachment at baseline is a major risk." True, but redundant, most AMD cases are of type 1 CNV, this is a vascularized PEDs, and most importantly it seems to be related to the size of the presenting PED and the stress on the RPE. Many times this complication occurs spontaneously, and relation to treatment is still controversial. It is suggested to exist but the extent of treatment contribution vs natural history on the event is debatable. Page 12, line 323: Macular atrophy definition and method of diagnosis differ between studies,  with study groups still trying to reach consensus on this. That is why it is not fully reported on RCTs and even if it was, differences in classification would make the data difficult to put together. This can be further/ better discussed in the manuscript. It is an important topic for ophthalmologists.

Author Response

Wednesday 06 May, 2020

Dear Reviewer 1,

       We appreciate the interest that you have taken in our manuscript and the constructive criticism you have given. Based on your comments, we have made changes to the manuscript, which are detailed below in italics. All modifications to the text in the manuscript are highlighted with red tracked changes.

Reviewer #1:

I have read with much interest the paper: "Comparative Safety of Bevacizumab, Ranibizumab, And Aflibercept For Treatment of Neovascular Age-Related Macular Degeneration (AMD): A Systematic Review and Network Meta-Analysis of Direct Comparative Studies", which I found clear and concise.   The main finding that the authors explore is the difference in adverse events among the 3 anti-VEGFs used in neovascular AMD treatment, a theme where much debate still exists despite multiple attempts to explore it, and an important one since it can change the perception and choice of treatments by physicians.   In fact, in the model created for comparison, it was found that "besides significant difference between RBZ and BVZ in systemic SAE rate no other significant difference revealed in either BVZ vs RBZ, RBZ vs AFB or RBZ vs AFB comparison." Which is an interesting and important finding but not totally unexpected. Also, by using p score (SUCRA analogue in frequentist network model) the authors found that "RBZ seems to have the potential of being superior over BVZ and AFB in terms of general surrogate safety parameters (dropout, death), over AFL in terms of CVD death, over BVZ in terms of systemic SAE." This is important information, but caution must always be kept in mind and further meta-analysis on the matter with real-life world data should be conducted to replicate these findings, as the authors correctly state in Conclusions.

Authors’ response:

Thank you for your positive feedback and valuable comments. We have now made changes to the manuscript according to your recommendations below.

  1. Introduction: Some minor errors and better English rephrasing is needed. Page 2 - contest instead of context.

Authors’ response:

Thank you for noticing that typo, we have substituted “contest” with “context” (line 60 in “all markups” view in Word).

  1. Page 3: "As regards the efficacy, there are a lot of RCT studies", perhaps "there are several, ".

Authors’ response:

Absolutely agree, thank you for your suggestion, we have accepted “there are several” wording (line 62 in “all markups” view in Word).

  1. Page 3: "Tear of retinal pigment and geographical atrophy", better describe those as Retinal pigment epithelium tear and geographic atrophy. However, the term "macular atrophy" would be preferable to use instead of "geographic/ geographical", as the latter is now used by consensus only in cases of cRORA (atrophy) but without a history of neovascularization. Same in page 5, line 135.

Authors’ response:

Thank you for this recommendation regarding unified terminology, so we have substituted both “Tear of retinal pigment” and “geographical atrophy” terms with “Retinal pigment epithelium tear (RPE tear)” and “macular atrophy” accordingly (lines 85, 86, 141, 337, 342 in “all markups” view in Word and in the Table 4).

  1. Also, in the introduction, the anti-VEGF discussion should be more focused on AMD only and comparative trials, and not extend to diabetic macular edema or venous occlusion, as the pathophysiology is entirely different compared to AMD and is out of the scope of the paper.

Authors’ response:

Fair comment, so we have made introduction a bit shorter (and we hope more readable as well) by removing statements and relevant references related to diabetic macular edema or venous occlusion (lines 64, 69-77 in “all markups” view in Word).

  1. Methods are appropriate, only minor questions to elucidate:

Page 4 line 121 - "safety of the following interventions: 1.25 mg BVZ, 0.5 mg RBZ, 2 mg AFB 121 every 2 months after 3 initial monthly doses (2q8), were included for network meta-analysis (NMA)." What were the interventions on BVZ and RBZ? Only AFB is described.

Authors’ response:

Thank you for this suggestion, we enriched and clarified intervention regimens in scope for BVZ, RBZ [monthly/as needed/treat-and-extend regimens] and also AFB [every 2 months after 3 months/treat-and-extend regimen] (lines 126-128 in “all markups” view in Word).

  1. Page 5, line 143: "events were presented as count and percentage.", perhaps change to frequencies and percentages.

Authors’ response:

Absolutely agree, we have substituted “count and percentage” with “frequencies and percentages” as recommended (lines 153 in “all markups” view in Word).

  1. Results: Table 4 - some results "Y" are in color red.  

Authors’ response:

Thanks for noticing that. It was technical mistake. Additionally, we changed “Y” to “N” in two relevant cells since BRAMD and Gefal studies didn’t report pseudoendophthalmitis.

  1. It would be nice to have a table exposing and detailing the main "systemic adverse" events explored in the included studies, as this is the outcome most important.  

Authors’ response:

Thank you for your attention to this point. We faced this potential source of heterogeneity while assessing sources of bias. The matter here is that “systemic serious adverse event (SAE)” has a clear definition according to Good Clinical Practices (ICH E2A) so it 1) results in death or life-threatening, 2) requires hospitalization or prolongation of hospitalization, 3) results in disability/incapacity, 4) Is a congenital anomaly/birth defect or 5) it is a medical situation requiring intervention to prevent one of the outcomes listed above. This definition is usually declared and supposed to be recognized homogenously in studies. At the same time, we revealed heterogeneity in reporting systemic SAE in several studies while assessing potential risk of bias (Section “Incomplete outcome data” Suppl.2), so we added additional statement regarding this fact referring to the relevant section in Supplement 2 (lines 147-151 in “all markups” view in Word). We would have developed a table you kindly proposed, however due to the variable approach in reporting across studies it is not seen feasible.

  1. Discussion: Page 11, line 276: "For instance, Zhang and co-workers (ref?) meta-analysis", is sentence meant to be like this? "(ref?)" Page 11, line 292: reference 22 in manuscript, but in the bibliography is ref 23 the real-life study.

Authors’ response:

Thanks, correct reference (Zhang et al, International Journal of Ophthalmology. 2014; 7(2):355–64) was missed due to technical issue, so we added it (line 289  in “all markups” view in Word) as reference #17, reorganized accordingly and double-checked all the other references.

  1. Page 12, line 319: " Several authors showed that almost all cases of RPE tear identified after anti-VEGF therapy was associated with a baseline vascularized PED. So vascularized pigment epithelial detachment at baseline is a major risk." True, but redundant, most AMD cases are of type 1 CNV, this is a vascularized PEDs, and most importantly it seems to be related to the size of the presenting PED and the stress on the RPE. Many times this complication occurs spontaneously, and relation to treatment is still controversial. It is suggested to exist but the extent of treatment contribution vs natural history on the event is debatable.

Authors’ response:

Agree, we changed this part of discussion according to your suggestion and clarified that our understanding of RPE tear nature and its relationship with anti-VEGF treatment is still controversial and supported this part of discussion with an additional reference: Gupta et al, Delhi J Ophthalmol 2017;27;243-9. (lines 329-336 in “all markups” view in Word)

  1. Page 12, line 323: Macular atrophy definition and method of diagnosis differ between studies, with study groups still trying to reach consensus on this. That is why it is not fully reported on RCTs and even if it was, differences in classification would make the data difficult to put together. This can be further/ better discussed in the manuscript. It is an important topic for ophthalmologists.

Authors’ response:

Thanks for this contribution, we enriched discussion section accordingly to reflect absence of standardized diagnosis definition as a possible reason for heterogeneity in macular atrophy reporting (lines 341-347 in “all markups” view in Word).

Looking forward to hearing from you.

Sincerely yours,

All coauthors

Reviewer 2 Report

In this study, authors reviewed and summarized safety results of previous studies.

Based on these data, systemic and ocular safety was compared among the three most commonly used anti-VEGF agents.

The study is well-written. In addition, the authors have excellently discussed their findings.

I have only few comments.

1. Page 4, lin2 128 ‘…two reviewers.’

Please indicate initials of the reviewers who performed the analysis.

2. One strong point of this study is that the ocular safety, including endophthalmitis, macular atrophy, and retinal pigment epithelial tear was compared among the different anti-VEGF agents.

However, the reason why analyzing these ocular adverse events is important was not presented.

I recommend to add paragraphs emphasizing the clinical significance of endophthalmitis, macular atrophy, and retinal pigment epithelial tear.

Author Response

Wednesday 06 May, 2020

Dear Reviewer 2,

     We appreciate the interest that you have taken in our manuscript and the constructive criticism you have given. Based on your comments, we have made changes to the manuscript, which are detailed below in italics. All modifications to the text in the manuscript are highlighted with tracked changes.

Reviewer #2:

 In this study, authors reviewed and summarized safety results of previous studies.

Based on these data, systemic and ocular safety was compared among the three most commonly used anti-VEGF agents.

The study is well-written. In addition, the authors have excellently discussed their findings.

I have only few comments.

Authors’ response:

Thank you for your positive feedback and helpful suggestions. We have now made changes to the manuscript according to your comments.

  1. Page 4, lin2 128 ‘…two reviewers.’ Please indicate initials of the reviewers who performed the analysis.

Authors’ response:

We specified reviewers’ names as you recommended in section 2.2 and 2.4 (lines 106, 134 accordingly in “all markups” view in Word)

  1. One strong point of this study is that the ocular safety, including endophthalmitis, macular atrophy, and retinal pigment epithelial tear was compared among the different anti-VEGF agents.

However, the reason why analyzing these ocular adverse events is important was not presented.

I recommend to add paragraphs emphasizing the clinical significance of endophthalmitis, macular atrophy, and retinal pigment epithelial tear.

Authors’ response:

Thanks for paying attention to this point. Among safety events we were specially interested in analyzing retinal pigment epithelium (RPE) tear and macular atrophy events since both of them are serious, may be observed during anti-VEGF treatment, considered at least partially associated with the treatment itself, potentially leading to the treatment withdrawal (which is often a disputable decision). We also highlighted endophthalmitis as one of the most dangerous local complication associated to intravitreal injections. So, we added appropriate explanation to the Introduction section (lines 85-91 in “all markups” view in Word).

Looking forward to hearing from you.

Sincerely yours,

All coauthors
